# Real-Time Monitoring of Dough Quality in a Dough Mixer Based on Current Change

**DOI:** 10.3390/foods13030504

**Published:** 2024-02-05

**Authors:** Wei Wang, Xiaoling Zhou, Wenlong Li, Jing Liang, Xiaowei Huang, Zhihua Li, Xinai Zhang, Xiaobo Zou, Bin Xu, Jiyong Shi

**Affiliations:** 1School of Food and Biological Engineering, Jiangsu University, Zhenjiang 212013, China; 2Chen Ke Ming Food Manufacturing Co., Ltd., Changsha 410000, China; 3International Joint Research Laboratory of Intelligent Agriculture and Agri-Products Processing, Jiangsu University, Zhenjiang 212013, China; 4College of Food Science and Engineering, Collaborative Innovation Center for Modern Grain Circulation and Safety, Nanjing University of Finance and Economics, Nanjing 210023, China

**Keywords:** dough, kneading, current, monitoring

## Abstract

Accurate assessment of dough kneading is pivotal in pasta processing, where both under-kneading and over-kneading can detrimentally impact dough quality. This study proposes an innovative approach utilizing a cost-effective current sensor to ascertain the optimal kneading time for dough. Throughout the kneading process, the dough’s tensile resistance gradually increases, reflecting the evolution of properties such as the gluten network. This leads to a discernible ascending phase in dough quality, evident through an increase in the load current of the mixing machine, succeeded by a subsequent decline beyond a certain threshold. The identification of this peak point enables the achievement of optimal dough consistency, thereby enhancing the overall quality of both the dough and subsequent pasta products. After the final product quality assessment, this novel method promises to be a valuable tool in optimizing pasta processing and ensuring consistent product quality.

## 1. Introduction

Dough products, revered as a globally cherished traditional food, enjoy widespread popularity. Central to the dough production process are the pivotal steps of mixing and resting, as elucidated by Obadi et al. [1]. Among these, dough kneading stands out as a critical phase with profound implications for both noodle quality and processing performance. This intricate procedure involves the uniform blending of flour and water, where flour components undergo hydration, giving rise to the gradual formation of gluten networks and the development of viscoelastic dough structures. Concurrently, air gas bubbles become entrapped in the dough matrix [2,3,4]. As expounded by Della Valle et al. and He et al. [5,6], the adequacy of the mixing process exerts a profound influence on the characteristics of the dough and, consequently, the quality of the end product. Insufficient mixing may result in inadequate amalgamation of wheat flour and water, insufficient gluten formation, and the presence of large starch particles not fully integrated into the dough. Conversely, over-mixing poses the risk of excessive gluten development, potentially leading to damage to the gluten structure of the dough [7]. Hence, meticulous control and monitoring of the dough mixing process become exceptionally imperative [3,8].

Monitoring of the dough preparation stage is an important step in the pasta production process [9,10]. The intricate interplay of various variables, encompassing dough texture parameters, the formation of the gluten network structure, and rheological properties, profoundly influences dough development [11]. To date, commonly employed methods for dough monitoring encompass visual inspection, texture testing, and rheology testing. Visual inspection entails a subjective evaluation of surface uniformity, smoothness, and brightness by an experienced observer. However, its inherently subjective nature poses limitations. Texture testing, utilizing a food substance meter to assess textural properties, lacks real-time online monitoring capabilities. Similarly, while rheological tests offer the ability to measure rheological, hardness, or textural characteristics of the dough [5,12], their accuracy is often contingent upon the specific dough recipe. Notably, prediction results can exhibit significant errors when applied to different recipes, precluding the feasibility of online real-time detection in the actual production process.

Mixers play a crucial role in the efficient mixing of dough for bread and pasta production, providing time savings and ensuring a more consistent dough texture compared to traditional hand mixing methods [13,14]. The capability for real-time monitoring of dough mixing progress is facilitated by the consistent process parameters offered by these mixers. In line with these advancements, our study introduces an innovative methodology that capitalizes on current changes during the flour mixing process to achieve optimal dough mixing effects. The dynamic nature of the mixer’s load, intricately linked with the tensile strength of the dough, is a focal point of our investigation. Changes in the dough’s rheological properties, encompassing factors such as stiffness, elasticity, and viscosity, alongside variations in the gluten network, moisture distribution, and microstructure, collectively influence its tensile resistance [15]. Our study takes a sophisticated approach by analyzing peak current variations, which directly correlate with both the mechanical and chemical properties of the dough. Through this comprehensive analysis, we can accurately predict the optimal mixing phase, promising not only improved dough processing efficiency but also elevated product quality.

## 2. Materials and Methods

### 2.1. Materials

The flour used in this experiment was high-gluten, medium-gluten, and low-gluten flour produced by Guchuan Flour (Beijing Guchuan Food Co., Beijing, China). High-gluten flour, with a protein content of 14% and an alveographic W value of 300, is compared to medium-gluten flour, which has a protein content of 11% and an alveographic W value of 130. Additionally, low-gluten flour features a protein content of 9% and an alveographic W value of 90. The dough mixer used was a KONKA multifunctional dough mixer with a rated voltage of 220 V; the mixer has dimensions of 320 mm in length, 185 mm in width, and 280 mm in height, with a rated power of 1200 w. The mixing hook is uniquely shaped in an ‘S’ configuration, measuring 11 cm in length and 8.5 cm in width.

### 2.2. Preparation of the Dough

The dough-making protocol employed in this study is an adaptation of the method outlined by Shen [16]. High-, medium-, and low-gluten flours, each weighing 1000 g, were individually combined with 350 g, 400 g, and 450 g of water, respectively. Subsequently, the mixtures were meticulously transferred to a precision mixer for the initial mixing phase. Following the initial mixing, the dough samples underwent further refinement in a blender. The mixing parameters and durations applied in the dough machine were as follows: an initial stir at the first speed (85 RPM) for 1 min, succeeded by continuous stirring at the second speed (135 RPM) for a duration of 10 min. This meticulous approach ensured a comprehensive and uniform amalgamation of the ingredients, aligning with the aims of our investigation. To capture the dynamic changes in the dough during the mixing process, dough samples were extracted at one-minute intervals. Each sampled portion was promptly sealed in an impermeable bag to forestall any moisture loss, maintaining the integrity of the samples for subsequent analyses.

### 2.3. Tensile Properties of the Dough

The tensile strength of the dough was meticulously examined at various stages of the mixing process. Dough samples were systematically collected at 2 min intervals, and dough strips were prepared for the analysis. The dough stretching experiments were conducted using a TA-XT2i Texture Analyzer (Stable Micro Systems Ltd., Godalming, UK), ensuring precision and reliability in the assessment. The test tape dimensions were standardized at 40 mm in length and 10 mm in width. Notably, the parameters for the dough texture analysis were fine-tuned based on insights from Zhang [17], ensuring an optimized and rigorous experimental setup. The tensile strength tests were executed in the “Return to Start” mode, employing an A/KIE probe. The specific parameters included a pre-test speed of 2.0 mm/s, a test speed of 3 mm/s, and a post-test speed of 10 mm/s. The starting pitch for each test was fixed at 30 mm, ensuring consistency across all experiments. Each sample underwent the tensile strength test six times to account for variability. The obtained results were meticulously averaged to derive the final tensile strength of the dough, providing a robust and comprehensive evaluation of the dough’s mechanical properties.

### 2.4. Measurement of the Change in Current in the Blending Process

The sensor used for current measurement in this study is an Asmik AC current transmitter with 0–10 A AC input. Current data were displayed and stored using Asmik’s industrial-grade single-channel paperless recorder with a sampling frequency set at 1 time/second. The hot wire of the stirrer power supply is passed through the measurement hole of the AC current transmitter, which is used to accurately measure the current data of the stirrer in real time. The changes in current corresponding to each stage of dough mixing are recorded.

### 2.5. Low Field Nuclear Magnetic Resonance (LF-NMR)

An NMI20-Analyst LF-NMR (Niumag Analytical Instrument Corporation, Suzhou, China) was used. Place the prepared dough in a 25-mm-diameter NMR test tube, scanning with the Carr-Purcell-Meiboom-Gill (CPMG) pulse sequence, and the number of sampling points is set to 10,004 (sampling frequency of 333.333 kHz). The sampling interval is set to 1500 ms, number of echoes is 4000, echo time is 0.1 ms, and number of accumulations is NS = 16 [18]. The inverse spectrum of the relaxation time T2 of the dough sample was obtained through a meticulous inversion procedure. The resulting spectrum revealed different crests, each corresponding to distinct forms of moisture within the dough. The peak area of each wave in the spectrum, expressed as a percentage of the total peaks, serves as a quantitative measure of the relative content of the respective forms of water present in the dough. This quantitative analysis provides valuable insights into the distribution and characteristics of different forms of moisture in the dough.

### 2.6. Dough Texture Analysis

Analyze the texture characteristics of the dough using the TA-XT2i Texture Analyzer. The dough was sampled and compressed for the analysis every minute using a 6-cm-diameter circular cutter to test its texture characteristics. The instrument was set in uncompressed mode, using a P/50 probe with the trigger type set to auto 10 g; before, during, and after testing at speeds of 2.0 mm/s, 0.5 mm/s, and 2.0 mm/s, respectively; 50% strain and 10 s between compressions. Six sheet dough samples were cut for each sample [19].

### 2.7. Confocal Laser Scanning Microscopy (CLSM) Analysis

The gluten mesh distribution of the dough can be captured through laser confocal microscopy. Based on methods by Zhang [2], the samples were cryo-embedded using a Leica tissue freezing medium and then cut into 20 μm sections using a cryosectioner. The dough sections were stained on slides for 1 min using a 0.025% rhodamine B reagent; then, the material was rinsed with demineralized water to remove rhodamine B from the surface, and coverslips were mounted. CLSM images were acquired at 40×, 1024 × 1024 pixels and 100 Hz were acquired, and the excitation and emission wavelengths of rhodamine B were set to 568 nm and 625 nm. Three samples of dough were prepared, and three prepared dough samples were selected at three different positions for image acquisition. The obtained images were analyzed and processed.

### 2.8. Scanning Electron Microscopy (SEM) Analysis

Samples of the dough were taken at 1 min intervals, and the dough samples before and after the maximum current were finally selected for the analysis. The three dough samples made were stationary in a 2.5% glutaraldehyde solution for 12 h. After 12 h, the dough samples were removed, and rinsed with a 0.1 M cold phosphate buffer solution; the rinsed samples were then freeze-dried and the sample surfaces were uniformly sprayed with gold particles. All samples were photographed with SEM at a 300 and 1000× magnification and 5 KV accelerating voltage. Finally, six sites were randomly selected for image acquisition for each sample, and they were analyzed and processed.

### 2.9. Final Noodle Quality

To determine the optimal cooking time for noodles, a set of 30 noodle samples, each subjected to varying mixing times, were individually immersed in 1000 mL of boiling water. Every 10 s, a single noodle was removed, gently compressed between glass sheets, and the point at which the disappearance of the central white core was observed was identified as the optimal cooking time.

Subsequently, the cooled noodle soup was transferred to a beaker and heated on an electric stove until boiling. Continuous stirring was maintained throughout the boiling process. As the majority of the water in the beaker evaporated, the beaker was then placed in an oven and dried to a constant weight. The slim cooking loss rate was calculated using the following formula:(1)$$L=\frac{Sd}{Nd}$$
where *L* stands for the cooking loss rate (%), *Sd* stands for noodle soup dry weight, and *Nd* stands for slim dry weight.

### 2.10. Statistical Analysis

Significant differences in the experimental results were assessed using non-parametric tests due to the assumption of non-normality. The Kruskal–Wallis test, followed by Dunn’s post hoc test for multiple comparisons, was employed for the data analysis. The experimental result data were processed using SPSS 26.0. Statistical significance was considered at a level of *p* < 0.05.

## 3. Results and Discussion

### 3.1. Analysis of the Tensile Capacity of the Dough

Tensile testing stands as a pivotal method for evaluating and refining dough performance, constituting a reliable approach in product assessment [20]. In the realm of dough mixing, the quantification of tensile properties centers on the dough’s resistance to the mixing process, specifically, its interaction with the dough mixer during kneading. This resistance manifests as the mixing resistance of the dough mixer, influencing critical parameters such as motor torque, power consumption, and load current of the dough mixing machine. Figure 1 elucidates the intricate relationship between different stages of dough mixing and the corresponding tensile resistance. Our investigation exposes a dynamic evolution in the tensile strength of the dough throughout the mixing process. Notably, the tensile strength reaches its zenith at the 8 min mark, showcasing a critical phase in the dough’s mechanical transformation, followed by a gradual decline thereafter. A noteworthy outcome of our study is the establishment of a method to translate changes in the dough’s tensile resistance to the corresponding load current of the dough mixing machine. This innovative approach facilitates real-time monitoring of dough kneading at each mixing stage, providing unprecedented insights into the mechanical dynamics of the process.

### 3.2. Measurement of Motor Current

In this study, commonly used current transmitters were employed as current sensors to measure the current in the mixer during the dough mixing process. Figure 2 illustrates the relationship between the current variation during dough kneading and the progress of dough making for low- and high-gluten flours with different water addition formulations. Figure 2a presents the current changes for low-gluten flour with varying water additions, while Figure 2b demonstrates the current changes for high-gluten flours with different water additions.

The magnitude of the current increases as the dough making process progresses due to the heightened resistance encountered during dough stretching. This increased resistance prompts a corresponding rise in the torque of the mixer motor, consequently leading to changes in the load current of the dough mixing machine. During the initial stages of dough mixing when the dough is in a flour-like state, the mixer torque and current are relatively low. Subsequently, as the dough undergoes the transition from a flour-like state to a flocculent state and then to a dough-like state, the tensile resistance gradually intensifies, resulting in amplified moisture content and interaction forces within the dough. Consequently, the motor load current of the dough machine noticeably increases.

Once a certain level is reached, the mixer’s current peaks alongside the maximum mixing resistance and dough stretching resistance. Thereafter, the current gradually levels off. In the subsequent mixing process, the dough becomes damaged due to over-mixing, leading to a decline in its tensile resistance and a subsequent decrease in the mixer’s current [21]. These findings indicate that the mixer’s current exhibits evident trends with variations in the dough development time during the kneading process across different formulations, hence enabling monitoring of the kneading process for various formulations.

### 3.3. Dough Moisture Distribution

LF-NMR uses the spin relaxation property of hydrogen protons in a magnetic field to analyze the form, distribution, and conversion of water in substances through the change in relaxation time [22]. Three optimal states of water exist in dough: bound water, free water, and semi-free and semi-bound water. The bound water tightly associates with starch or gluten proteins. Free water exhibits the highest fluidity. Semi-free and semi-bound water exists in an intermediate state, forming connections with macromolecules such as proteins and starches. This classification delineates the varying hydration states crucial for understanding the dough’s structural and textural properties. The maximum tensile resistance of the dough is positively correlated with the relaxation time and the amount of deeply bound water; the lower the free water, the better the processing of the dough [23]. Moreover, the stabilization of the dough proceeds with the migration of weakly bound water to bound water, there was a more structured secondary structure of dough protein, and the microstructure of the dough was more homogeneous [24]. Therefore, the form and distribution of water can effectively reflect the characteristics of the workmanship quality of the dough [25], and the workmanship quality characteristics of the dough can be effectively reflected by the morphology and distribution of water.

As demonstrated in Figure 3 and Table 1, the moisture content in the dough sample primarily consisted of weakly bound water, with A22 accounting for more than 80% of the total water content. Notably, the presence of deeply bound water, which significantly impacts the gluten network structure, was relatively low, potentially contributing to the dough’s lower tensile strength. Conversely, the high proportion of weakly bound water resulted in increased stickiness of the oat dough flakes. Comparing the moments before and after the maximum current moment to the moment of the maximum current, it was observed that the proportion of deeply bound water decreased. This reduction could be attributed to inadequate mixing, hindering the complete binding of gluten to water molecules. On the other hand, over-mixing led to the disruption of the gluten network and the release of some water previously bound to gluten. The migration of water within the dough strip elucidated the influence of stirring action on the dough’s stretching ability. These findings highlight the importance of water distribution and its impact on the workmanship quality characteristics of the dough.

### 3.4. Dough Texture Analysis (TPA)

The structural properties of the dough are largely influenced by the mixing time, and the dough exhibits different structural properties at different mixing stages [26]. The quality characteristics of the flaky dough with 35% water in the mixer and different currents in the mixer are shown in Table 2. The flaky dough with the highest current loading time of the mixer had the highest firmness, elasticity, adhesion, cohesiveness, gluing, chewiness, and resilience in all time periods.

The intensity of the gluten network and the forces within the dough vary during different stages of the dough kneading process. Initially, as the current gradually increased, the qualitative properties of the dough such as hardness and stickiness were suboptimal. However, these properties, along with elasticity, adhesiveness, and chewiness, significantly improved when the load current reached its maximum level. Upon reaching the maximum current, the dough exhibited optimal hardness, viscosity, elasticity, stickiness, and chewiness. Subsequently, these parameters showed a decreasing trend. Prolonged mixing times resulted in a significant decrease in hardness, stickiness, and chewiness as the mixer current started to decrease. Conclusively, the quality characteristics of the dough undergo changes throughout the kneading process, impacting the mixer load. The results indicate that the dough performs best when the mixer load current reaches its maximum level.

### 3.5. Confocal Laser Scanning Microscopy (CLSM) Analysis

CLSM is a method for exploring gluten network changes through physical visualization of major dough structural factors [25]. Figure 4 shows the original and fitted CLSM images of the tissue network distribution during the kneading of the database dough. As the current in the mixer reaches its maximum, the overall gluten structure of the density in the dough becomes well developed and forms a coherent network of gluten. The starch pellets are immersed evenly in the bran substrate and arranged closely and orderly with the bran. For under-mixed and over-mixed doughs (mixer current less than maximum current), the network of bran showed a loose structure and a non-uniform distribution of the bran network. This could be the outcome of an underdeveloped gluten network in the case of under-mixing and damage due to disruption of the gluten structure in the case of over-mixing. To further investigate the transformation of structured gluten in the network of dough with different kneading and resting times, CLSM images were processed. Table 3 summarizes the gluten percentage area, gluten connections, total gluten length, average gluten length, cracking, and branching rate. For fully stirred dough, gluten percentage area, gluten connections, total gluten length, average gluten length, and branching rate showed more significant increases than those of under-stirred dough, and under-stirred dough showed more Lacunarity. For over-stirred dough, there was a significantly lower branching rate and average. This is because excessive mixing of the dough disrupts the network of gluten, and the length of the gluten mesh decreases and Lacunarity increases. The CLSM results also explain some of the characteristics of the dough TPA and the variation pattern of the load current during dough mixing. A good gluten structure directly influences the high product quality and the success of the subsequent dough.

### 3.6. Scanning Electron Microscopy (SEM)

Scanning electron microscopy is well positioned to investigate the microstructural development of mixed doughs [27]. Scanning electron microscopy was used for observing the changes in the dough microstructure at different mixer currents during the mixing process with magnifications of 300× and 600×. During all mixing processes, the dough showed a relatively contiguous network of gluten, with starch particles tightly embedded in the gluten network, and fewer pores and cracks at the stage when the mixer current reached its maximum current, as shown in Figure 5c,d. Under insufficient stirring with the stirrer current less than the maximum current as in Figure 5a,b, the gluten network appears discontinuous and most of the amylose particles cannot be fully embedded in the gluten network. Under excessive agitation, as in Figure 5e,f, the gluten network was disrupted and cracks appeared. This result indicates that proper mixing plays a key role in the evolution and formation of the structure of the flour gluten, and over-mixing or under-mixing can affect the development of the dough and ultimately the subsequent processing and cooking of the dough products [28]. This discovery indicates that alterations in the loading current of the dough mill can accurately mirror the progression of dough microstructure development.

### 3.7. Comparison of Methods for Determining the Optimal Mixing Time

To investigate the advantages of using the current to monitor the kneading progress of the dough to determine the optimal dough over determining the optimal dough through mixing time, important parameters such as texture, zone network condition, and moisture distribution were selected for comparison. According to Liu, the optimal kneading time of the dough was 8 min [7]. The dough samples obtained with a mixing time of 8 min and the moment of the maximum current were selected for comparison during one dough mixing process. Table 4 shows the results of comparing the dough with a mixing time of 8 min to the dough with the maximum current moment during mixing. The results show that the dough obtained at the maximum load current of the mixer motor was better than the dough at the 8 min moment in terms of texture characteristics, gluten mesh area, total length, average length, etc., and moisture distribution. Therefore, the use of current sensors to monitor the kneading process has advantages over conventional methods and can be used to improve the efficiency and quality of the kneading process and ensure the production of high-quality dough products.

### 3.8. The Cooking Quality of Noodles

Table 5 presents crucial data on the optimal cooking time and cooking loss rate of noodles. The optimal cooking time signifies the duration required for noodles to achieve full cooking, offering insights into the extent of dough development and firmness. In general, a lengthier cooking time corresponds to a finer dough structure, reflecting the formation of the gluten network. Notably, the dough extracted at the point of the maximum current exhibits a prolonged optimal cooking time compared to other noodles. Conversely, dough of a lower quality, characterized by incomplete internal development, demonstrates faster water entry during cooking, resulting in a shorter cooking time.

The cooking loss rate pertains to the concentration of dissolved substances in water during the cooking process, and a high cooking loss rate can adversely affect the palatability of noodles. Insufficient gluten development in the dough leads to the detachment of starch particles during cooking, consequently amplifying the cooking loss. This phenomenon significantly impacts the overall quality of the cooked noodles. The experimental results demonstrate the effectiveness of monitoring dough development through the electric current.

## 4. Conclusions

During dough mixing and processing, the dough undergoes changes in its tensile properties, textural properties, moisture distribution, gluten network structure, and microstructure. Obtaining the best performing dough with optimal performance requires careful monitoring and control of the mixing and stirring processes. Real-time monitoring and control of these processes in food processing present challenges. However, by translating the changes occurring in the mixing process into easily monitored variables, it becomes feasible to achieve effective control. This paper introduces a method that converts the degree of dough mixing into variations in the mixer load current. By measuring the change in mechanical load through the traction current of the mixer motor, the readiness of the dough processing can be determined. The proposed monitoring method offers a cost-effective approach to automating various production equipment, which has significant implications for the dough processing process. This technique facilitates timely adjustments and ensures the production of high-quality dough. The application of this monitoring method holds promise for enhancing the efficiency and quality control of dough production in the food industry. By implementing real-time measurements of the mixer load current, operators can optimize the dough processing and improve overall production processes. This advancement in automation provides valuable assistance in achieving consistent and high-quality results in dough processing.

## Figures and Tables

**Figure 1 foods-13-00504-f001:**
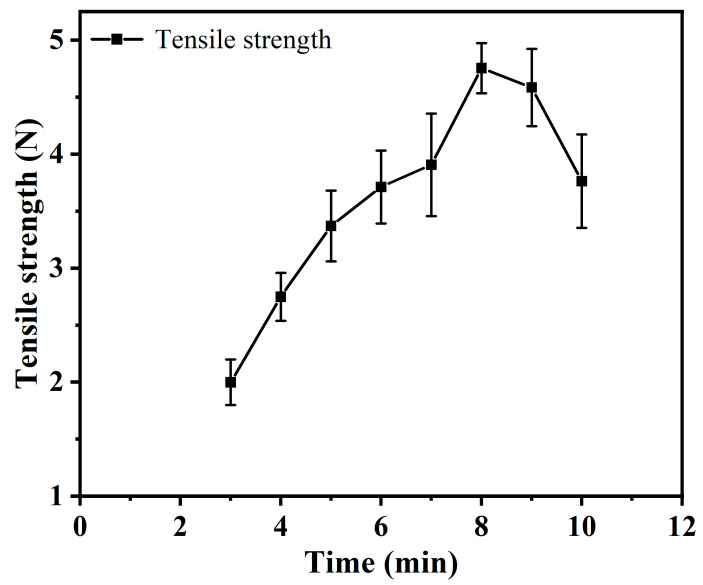
Variation of tensile capacity of the face tape with time.

**Figure 2 foods-13-00504-f002:**
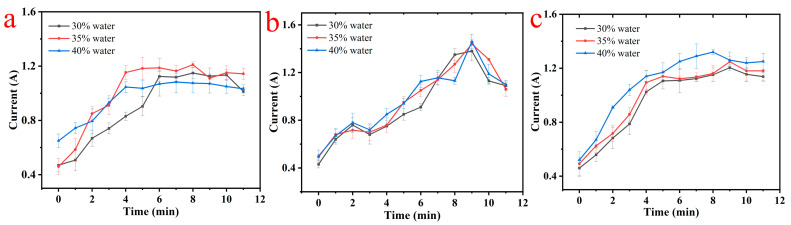
(**a**) Mixing current and mixing time curves for high-gluten flour at different water addition amounts; (**b**) mixing current and mixing time curves for medium-gluten flours at different water additions; (**c**) mixing current and mixing time curves for low-gluten flours at different water additions.

**Figure 3 foods-13-00504-f003:**
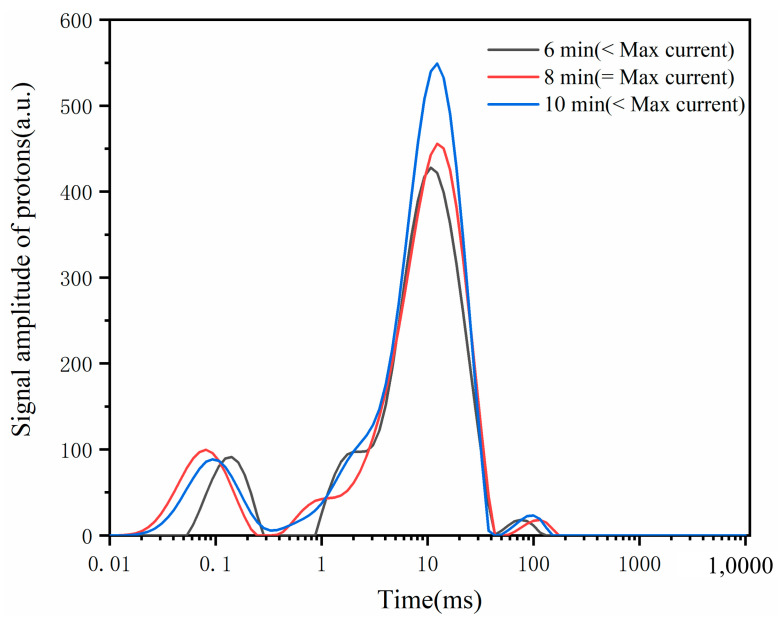
Spin–spin relaxation time of dough at different currents.

**Figure 4 foods-13-00504-f004:**
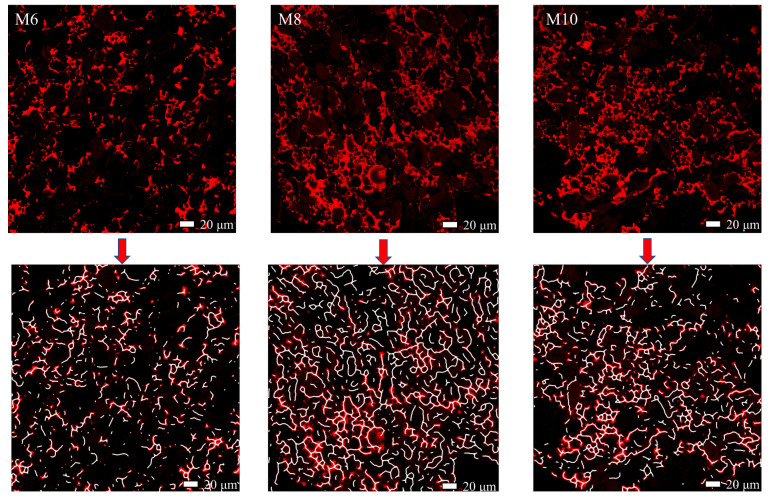
Before and after images of gluten network processing. M6, M8, M10 indicate mixing for 4 min, 8 min, 10 min. Red indicates gluten network, and white indicates gluten network skeleton.

**Figure 5 foods-13-00504-f005:**
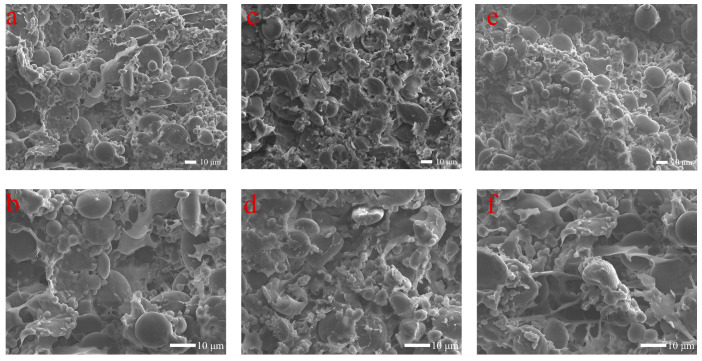
Scanning electron micrographs of cross-sections of flaky dough at different time stages after mixing in the dough mixing machine: <Max. current (=6 min) ((**a**): ×600, (**b**): ×1000), =Max. current (=8 min) ((**c**): ×600, (**d**): ×1000), <Max. current (=10 min) ((**e**): ×600, (**f**): ×1000).

**Table 1 foods-13-00504-t001:** Moisture state and distribution of dough at different current stages.

	<Max. Current (=6 min)	=Max. Current (=8 min)	<Max. Current (=10 min)
T_21_ (ms)	0.22 ± 0.01 ^c^	0.26 ± 0.06 ^a^	0.23 ± 0.01 ^b^
T_22_ (ms)	42.15 ± 0.54 ^b^	45.03 ± 0.36 ^a^	42.96 ± 0.01 ^b^
T_23_ (ms)	88.59 ± 3.71 ^c^	117.40 ± 1.30 ^a^	108.70 ± 0.00 ^b^
A_21_ (%)	10.75 ± 0.70 ^c^	14.25 ± 0.51 ^a^	12.01 ± 1.01 ^b^
A_22_ (%)	88.07 ± 0.5 ^a^	84.67 ± 0.64 ^c^	86.70 ± 0.24 ^b^
A_23_ (%)	1.18 ± 0.20 ^b^	1.08 ± 0.18 ^c^	1.29 ± 0.17 ^a^

Results are expressed as mean ± standard deviation, and different lowercase letters in the same line indicate significant differences (*p* < 0.05).

**Table 2 foods-13-00504-t002:** Textural properties of different current phases.

	<Max. Current (=6 min)	=Max. Current (=8 min)	<Max. Current (=10 min)
hardness	3579.12 ± 700 ^b^	4109.18 ± 975 ^a^	3025.98 ± 262 ^c^
adhesiveness	2550.59 ± 211 ^a^	5567.25 ± 210 ^a^	3261.14 ± 369 ^b^
resilience	8.50 ± 1.28 ^b^	10.08 ± 2.33 ^a^	8.88 ± 0.80 ^b^
cohesion	0.29 ± 0.03 ^b^	0.61 ± 0.07 ^a^	0.57 ± 0.04 ^a^
springiness	19.00 ± 3.48 ^c^	84.66 ± 9.78 ^a^	55.17 ± 4.63 ^b^
gumminess	1036.79 ± 297 ^c^	2103.12 ± 197 ^a^	1725.77 ± 226 ^b^
chewiness	204.77 ± 97 ^c^	1564.31 ± 513 ^a^	1006.58 ± 117 ^b^

Results are expressed as mean ± standard deviation, and different lowercase letters in the same line indicate significant differences (*p* < 0.05).

**Table 3 foods-13-00504-t003:** Analysis of protein networks in mixed samples with Angio Tool software (version 0.6).

	<Max. Current (=6 min)	=Max. Current (=8 min)	>Max. Current (=10 min)
Gluten percentage area (%)	18.22 ± 3.08 ^c^	25.26 ± 2.27 ^a^	23.07 ± 3.48 ^b^
Gluten junctions	180.25 ± 55.77 ^c^	299.25 ± 55.71 ^b^	327.75 ± 52.0 ^a^
Total gluten length (×10^3^ μm)	15.88 ± 3.51 ^b^	21.06 ± 2.5 ^a^	21.56 ± 2.81 ^a^
Average gluten length (μm)	54.45 ± 0.88 ^c^	103.24 ± 2.21 ^a^	79.28 ± 2.72 ^b^
Lacunarity (×10^−2^)	28.73 ± 2.02 ^a^	22.02 ± 1.28 ^b^	22.64 ± 2.31 ^b^
Branching rate (×10^−3^)	1.95 ± 0.23 ^b^	2.28 ± 0.22 ^a^	1.96 ± 0.27 ^b^

Results are expressed as mean ± standard deviation, and different lowercase letters in the same line indicate significant differences (*p* < 0.05).

**Table 4 foods-13-00504-t004:** Comparison of dough parameters at eight minutes of mixing with dough at the moment of maximum current.

	8 Min	Maximum Current
Hardness	3551.63 ± 276 ^b^	4197.60 ± 188 ^a^
Adhesiveness	4109.14 ± 140 ^a^	4157.73 ± 134 ^a^
Resilience	11.36 ± 0.96 ^a^	12.73 ± 0.82 ^a^
Cohesion	0.58 ± 0.07 ^b^	0.69 ± 0.02 ^a^
Springiness	73.47 ± 2.48 ^b^	89.14 ± 11.11 ^a^
Gumminess	2458.65 ± 321 ^a^	2519.63 ± 43 ^a^
Chewiness	1664.38 ± 206 ^b^	1847.58 ± 76 ^a^
Gluten percentage area (%)	21.16 ± 2.29 ^b^	25.92 ± 2.42 ^a^
Gluten junctions	315.30 ± 12.83 ^b^	338.75 ± 34.06 ^a^
Total gluten length (×10^3^ μm)	17.42 ± 1.29 ^a^	18.20 ± 2.40 ^a^
Average gluten length (μm)	68.29 ± 10.74 ^b^	74.58 ± 7.34 ^a^
Lacunarity (×10^−2^)	26.79 ± 1.49 ^a^	28.73 ± 1.98 ^a^
Branching rate (×10^−3^)	1.86 ± 0.26 ^a^	2.13 ± 0.30 ^a^
A_21_ (%)	9.62 ± 1.18 ^b^	13.67 ± 0.82 ^a^
A_22_ (%)	89.23 ± 5.08 ^b^	85.12 ± 4.83 ^a^
A_23_ (%)	1.15 ± 0.14 ^a^	1.21 ± 0.19 ^a^

Results are expressed as mean ± standard deviation, and different lowercase letters in the same line indicate significant differences (*p* < 0.05).

**Table 5 foods-13-00504-t005:** The cooking quality of noodles.

	Optimum Cooking Time (s)	Cooking Loss Rate (%)
<Max. current	210	13.86 ± 0.34 ^a^
=Max. current	280	8.34 ± 0.49 ^c^
>Max. current	280	9.73 ± 0.56 ^b^

Results are expressed as mean ± standard deviation, and different lowercase letters in the same line indicate significant differences (*p* < 0.05).

## Data Availability

Data is contained within the article.

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
