# Peer review of "Real-Time Monitoring of Dough Quality in a Dough Mixer Based on Current Change"

_foods, 2024, doi:10.3390/foods13030504_

Round 1

Reviewer 1 Report

Comments and Suggestions for Authors

In the abstract, it was stated that the monitoring of dough quality was evaluated through the tensile resistance as an indication of gluten network, and this conclusion was connected to an optimal dough quality, and finally to the product quality. For this reason, a product evaluation was expected, but, only dough properties were discussed.

The manuscript presented different measurements of pasta dough prepared with flours of different gluten contents (in terms of low and high) and different amount of water added. However, without an investigation of the end products prepared with these dough samples, it is difficult to evaluate the significance of the proposed procedure.

In Table 4: What is the reason of taking subtraction of parameters at two conditions? For example, what is the significance of having resilience difference of 1.37?  Wouldn’t it be a more objective comparison with a Tukey analysis?

Fig.5  a, b, and f: the images are not clear.

In Fig.5, it is difficult to see the differences between X300 and X600 magnification. Is it possible to give X300 and X1000 or X600 and X1000?

Comments on the Quality of English Language

An editing of the language might be needed. For example:

Line 202. The sentence can be rephrased, such as “the existence of water in dough can be explained as deeply combined water, ……..”.

Lines 298-299. Please rephrase the sentence. "dough mixer can achieve the effect of monitoring" not clear.

Reviewer 2 Report

Comments and Suggestions for Authors

The article presents an analysis of current changes during the dough mixing process. It is shown that these changes are correlated with the mechanical and chemical properties of the dough. This correlation makes it possible to monitor and control the parameters of the dough production process. Thus, it makes it possible to optimize the parameters of the dough in terms of its production efficiency and food quality. The presented method makes it possible to monitor and control the dough mixing and manufacturing process in real (production) time. 

The article was correctly prepared from the methodological aspect, the planned research experiment, the analysis of the results and the discussion with the works of other authors. I think that the article is interesting from the scientific side, as well as the high probability of applying the results in real conditions.

 I accept the article in this form.

Author Response

Dear reviewers:

We are grateful for your valuable comments and suggestions, which helped to improve the quality of our manuscript entitled “Real-Time Monitoring of Dough Quality in Dough Mixer Based on Current Change” (No.: foods-2771881).

Reviewer 3 Report

Comments and Suggestions for Authors

General overview.

Manuscript foods-2771881- titled “Real-Time Monitoring of Dough Quality in Dough Mixer 2 Based on Current Change”, reported the possible application of different techniques to evaluate in real time the dough quality in dough mixer. The manuscript is very interesting however, some modifications should be made to clarify material and methods used and obtained results.

Major comment

Material and methods

More detailed samples description should be added. It would be interesting if the authors reported the centesimal composition of the flours and the alveographic W parameter.

Statistical data elaboration should be reviewed considering the sample size. I suggest to use non parametric test for variances evaluations. In addition more information about LF-NMR data elaborations should be added.

Result

Figure 1 I suggest to add the variation of all samples and the standard deviation

Figure 2 I suggest to add the curve of stirring current and stirring time of medium gluten flour samples

Line 201 In food matrix is not easy define the exact proton class. I think that you have three major class in which is possible summarizing the continues LF-NMR T2 signal distributions. Please add NMR T2 images or graphic with class distribution.

Figure 3 is not clear if referred to LF-NMR or what. And if referred to LF-NMR relaxation time I found more than 3 protonic class and time shifting in relation to the max current. Please revise the LF NMR section adding this information.

Discussion

Data discussions are very short. Please add more discussion taking in consideration the previously literature data.

Round 2

Reviewer 1 Report

Comments and Suggestions for Authors

Suggested and required changes were done.

Author Response

(The authors gave the same response as above.)

Reviewer 3 Report

Comments and Suggestions for Authors

No additional comment

Author Response

(The authors gave the same response as above.)
